# Nanostructured Surfaces to Promote Osteoblast Proliferation and Minimize Bacterial Adhesion on Titanium

**DOI:** 10.3390/ma14164357

**Published:** 2021-08-04

**Authors:** Samira Esteves Afonso Camargo, Xinyi Xia, Chaker Fares, Fan Ren, Shu-Min Hsu, Dragos Budei, Chairmandurai Aravindraja, Lakshmyya Kesavalu, Josephine F. Esquivel-Upshaw

**Affiliations:** 1Department of Restorative Dental Sciences, Division of Prosthodontics, University of Florida College of Dentistry, Gainesville, FL 32610, USA; scamargo@dental.ufl.edu (S.E.A.C.); shuminhsu@ufl.edu (S.-M.H.); 2Department of Chemical Engineering, Herbert Wertheim College of Engineering, University of Florida, Gainesville, FL 32611, USA; xiaxinyi@ufl.edu (X.X.); c.fares@ufl.edu (C.F.); fren@che.ufl.edu (F.R.); 3Dentix Millennium SRL, 087153 Giurgiu, Romania; budei.dragos@dentixmillennium.ro; 4Department of Periodontology, College of Dentistry, University of Florida, Gainesville, FL 32611, USA; AChairmandurai@dental.ufl.edu (C.A.); KESAVALU@dental.ufl.edu (L.K.)

**Keywords:** nanotubes, implant, osteoblasts, coating, SiC

## Abstract

The objective of this study was to investigate the potential of titanium nanotubes to promote the proliferation of human osteoblasts and to reduce monomicrobial biofilm adhesion. A secondary objective was to determine the effect of silicon carbide (SiC) on these nanostructured surfaces. Anodized titanium sheets with 100–150 nm nanotubes were either coated or not coated with SiC. After 24 h of osteoblast cultivation on the samples, cells were observed on all titanium sheets by SEM. In addition, the cytotoxicity was evaluated by CellTiter-BlueCell assay after 1, 3, and 7 days. The samples were also cultivated in culture medium with microorganisms incubated anaerobically with respective predominant periodontal bacteria viz. *Porphyromonas gingivalis*, *Treponema denticola*, and *Tannerella forsythia* as monoinfection at 37 °C for 30 days. The biofilm adhesion and coverage were evaluated through surface observation using Scanning Electron Microscopy (SEM). The results demonstrate that Ti nanostructured surfaces induced more cell proliferation after seven days. All groups presented no cytotoxic effects on human osteoblasts. In addition, SEM images illustrate that Ti nanostructured surfaces exhibited lower biofilm coverage compared to the reference samples. These results indicate that Ti nanotubes promoted osteoblasts proliferation and induced cell proliferation on the surface, compared with the controls. Ti nanotubes also reduced biofilm adhesion on titanium implant surfaces.

## 1. Introduction

Dental implants are becoming a common choice to replace lost teeth in many cases. However, these implants can be susceptible to peri-implantitis and peri-mucositis, which are diseases of the supporting structures of the implant [1,2,3]. The most effective treatment for peri-implantitis is prevention by minimizing bacterial colonization and implementing implant maintenance regimens [4]. Thus, the development of preventive strategies, such as implant surface modification, to minimize bacterial colonization and proliferation of the implant, are necessary [5].

Despite the recognized positive mechanical and chemical properties of titanium-based materials for orthopedic and dental applications, reduced osseointegration of the implants, bacterial attachment, and intense inflammatory response from the host pose significant problems that need to be answered [6,7].

Different techniques for metal biomaterials have been developed to modify their surfaces, studying a better biological response after their implantation [5,8]. Studies have shown that human cells are suitable for interacting with nanostructured surfaces, verifying that these surfaces can associate with some proteins more effectively than conventional materials [9,10,11].

TiO_2_ nanotubes are one of the promising bone/dental implant surface modification strategies with enhanced bioactivity and local therapeutic functions [12,13,14]. Recently, there has been an increased interest in investigating nanoscale surface topographies as biomimetic interfaces for implantable devices. [15], particularly in improving the osteoblast function and, also in antibacterial potential. The effects are largely due to the TiO_2_ nanotubes geometry and physicochemical properties [16].

Although TiO_2_ nanotubes demonstrate excellent corrosion resistance and biocompatibility [15,16], previous studies have shown that oral bacteria not only contribute to titanium corrosion but also can expedite the process [4,17,18]. Our previous research exhibited the effect of silicon carbide (SiC) in decreasing surface corrosion, fracture, and bacterial adhesion. Coating TiO_2_ nanotubes with SiC can potentiate the excellent characteristics of this nanostructure by preventing the deleterious effect of bacteria on the surface [4].

In recent years, the effectiveness of using TiO_2_ nanotube-containing surfaces to increase cell proliferation has been demonstrated [11,19]. Titania nanotubes (TNTs) with a diameter of 100 nm were able to decrease *Staphylococcus aureus* attachment and simultaneously improve the mesenchymal stem cell response [20].

Ti-based nanotubes enhance the surface roughness of the implant, increasing the osseointegration, antibacterial effects, and in addition, can be functionalized in many ways, opening a wide range of possibilities [21,22].

Therefore, it is important to have a dental implant that reduces the biofilm adhesion and promotes bone proliferation. The objective of this study was to investigate the potential of titanium nanotubes to promote the proliferation of human osteoblasts and to reduce monomicrobial biofilm adhesion. As *P. gingivalis*, *T. denticola,* and *T. forsythia* are the predominant oral subgingival bacteria that cause periodontitis and tend to form biofilm on the tooth supragingival and subgingival surface, these three oral bacteria were utilized for the present study. Further, these bacteria were strongly associated with peri-implantitis [1], which will provide an opportunity to study the anti-biofilm potential of Ti-based nanotubes on dental implants. A secondary objective was to determine the effect of silicon carbide (SiC) on these nanostructured surfaces.

## 2. Materials and Methods

### 2.1. Experimental Design

Forty-eight titanium sheets (10 × 10 mm^2^) and 24 titanium nanotube sheets (10 × 10 mm^2^, InRedox, Longmont, CO, USA) were used in this study. These titanium nanotube sheets are made through anodization of titanium oxide with nanotube diameters of 100 nm and 150 nm.

Three experimental groups (*n* = 12) were included as follows: titanium oxide silicon coated with silicon carbide (TiO_2_-SiC), titanium oxide silicon nanotubes 100 nm (TiO_2_ nano), and titanium oxide silicon nanotubes 150 nm coated with silicon carbide (TiO_2_ nano-SiC). The sizes of the nanotube diameters (100 nm and 150 nm) were chosen according to the dental implants from Dentix Millennium SRL (Giurgiu, Romania) (Figure 1).

### 2.2. Coating Process

Twenty-four titanium sheets were cleaned with acetone (ACS reagent > 99.5%, CAS#67-64-1, Sigma-Aldrich, St. Louis, MO, USA) in an ultrasonic bath, then rinsed with isopropyl alcohol (CAS#67-63-0, Sigma-Aldrich) and dried with compressed nitrogen. The sheets were then treated with ozone to remove any surface carbon contamination. Then, the titanium sheets were coated with silicon carbide (SiC). To coat samples with 200 nm of SiC, a plasma-enhanced chemical vapor deposition system (PECVD, PlasmaTherm 790, Saint Petersburg, FL, USA) was utilized. The gas precursors used for the SiC deposition were methane (CH_4_) and silane (SiH_4_) at a deposition temperature of 300 °C [23]. The dental implant 150 nm was also coated with SiC to show that it is possible to cover the surface (Figure 1B).

### 2.3. Bacterial Culture

To study the bacterial adhesion on the titanium sheets, predominant periodontal bacterial pathogens viz. *P. gingivalis* FDC 381, *T. denticola* ATCC 35405, and *T. forsythia* ATCC 43037 were used as monobacterial infection. *P. gingivalis* FDC 381 was grown in Brucella blood agar plate supplemented with hemin and vitamin K (Hardy Diagnostics, Santa Maria, CA, USA). The oral spirochete *T. denticola* ATCC 35405 was grown in GM-1 broth. *T. forsythia* was grown in tryptic soy (Soybean Casein Digest Medium, BD Bacto^TM^, Franklin Lakes, NJ, USA) broth supplemented with hemin (5 µg/mL, Sigma-Aldrich) and N-acetylmuramic acid (1 mg/mL, Sigma-Aldrich). All the bacteria were grown and maintained in Coy anaerobic chamber at 37 °C for 3 days, as previously described [24,25,26]. *P. gingivalis* was harvested from the media plates using sterile cotton tip applicator. The log-phase culture of *T. denticola* and *T. forsythia* were harvested by centrifugation (8000 rpm for 10 min) and the pellet was washed once with phosphate-buffered saline (PBS). Bacteria were suspended in sterile reduced transport medium (RTF) and vigorously vortexed. The axenic nature of the bacteria was assessed by Gram staining. The number of bacterial cells in the suspension was determined using the Petroff-Hausser bacterial counting chamber. Bacteria were diluted in RTF to reach the final concentrations of 10^10^ cells/mL for *P. gingivalis*, 2 × 10^8^ cells/mL for *T. denticola*, 3 × 10^8^ cells/mL for *T. forsythia* and all the tested implants were placed in 1 mL of bacterial suspension individually in 24-well sterile plates in an anaerobic chamber. The 24-well plates were maintained in anaerobic growth chamber for 30 days with respective fresh media replenishments every three days.

### 2.4. Biofilm Adhesion and Coverage

After 30 days, the biofilm adhesion on the Ti sheets surface were identified under scanning electron microscopy using a MAICE system (JEOL JSM-6400 Scanning Electron Microscope, JEOL LTD, Tokyo, Japan). The monomicrobial biofilm adhering to the samples was fixed in a solution of 3% glutaraldehyde (50 wt.% in H_2_O, CAS#111-30-8, Sigma-Aldrich), 0.1 mol/L sodium cacodylate (CAS#6131-99-3, Sigma-Aldrich), and 0.1 mol/L sucrose (CAS#57-50-1, Sigma-Aldrich) for 45 min. Samples were placed in a buffer solution of 0.1 mol/L sucrose (CAS#57-50-1, Sigma-Aldrich) and 0.1 mol/L sodium cacodylate (CAS#6131-99-3, Sigma-Aldrich) for 10 min. Ti sheets surface and biofilm were treated in serial ethanol dehydrations for 10 min each and dehydrated in hexamethyldisilazane (HDMS, CAS# 999-97-3, Sigma-Aldrich) until SEM imaging. Ti sheets were then sputter-coated with a palladium–gold alloy (Polaron SC 7620 Sputter Coater, Quorum Technologies, Laughton, East Sussex, UK) with a thickness of 10 nm to reduce charging effects during SEM analysis (10–15 mA, under a vacuum of 130 mTorr). After this, the SEM was operated at 5 kV, spot 3 to 6.

In addition, SEM images of the biofilm on the surface of the Ti sheets were analyzed by ImageJ software. Bacteria coverage percentages were averaged over five random areas.

### 2.5. Analysis of Nanotube Surface by SEM

Non-coated and coated nanotubes sheets were observed under scanning electron microscopy using a MAICE system (JEOL JSM-6400 Scanning Electron Microscope, JEOL LTD, Tokyo, Japan) to identify surface roughness after cultivation with a biofilm. Coated and non-coated nanotubes sheets were incubated for 30 days. The biofilm adhered to the samples was removed by placing the samples inside a Falcon tube washed 3× with 2 mL of distilled water, and sonicated for 5 min. The SEM was operated at 5 kV, spot 3 to 6 and the images were recorded.

### 2.6. Cell Analyses

For cell analyses, titanium sheets from each group were sterilized in autoclave (121 °C, 60 min) and placed on a sterile 24-well plate. Human Osteoblasts (Nhost, Lonza, Walkersville, MD, USA) were cultivated in polystyrene vented tissue-culture flasks (surface area 175 cm^2^) at 37 °C in 5% CO_2_ until confluency was reached. Dulbecco’s Modified Eagle Medium (DMEM, Sigma-Aldrich), supplemented with 10% fetal bovine serum and 1% penicillin/streptomycin, was used. The cells of passage 3 to 10 were used for subsequent experiments. A quantity of 20,000 cells/well were cultivated on the titanium sheets in 24-well plates at 37 °C in a 5% CO_2_ for 1 day.

#### 2.6.1. Cytotoxicity

Cell viability were determined by a CellTiter-BlueCell Viability Assay (Promega, Madison, WI, USA, G808A), which was used according to the manufacturer’s instructions. After 1, 3, and 7 days, 50 μL of CellTiter-Blue dye was added to coated and non-coated titanium sheets for every 500 μL of culture media, and samples were incubated for 4 h at 37 °C and 5% CO_2_. Sample fluorescence was analyzed using a spectrophotometer (SmartSpec Plus, Bio-Rad, Hercules, CA, USA) at a wavelength of 600 nm, which generated density optic values.

#### 2.6.2. Cell Attachment

Titanium sheets were observed under scanning electron microscopy using a MAICE system (JEOL JSM-6400 Scanning Electron Microscope, JEOL LTD, Tokyo, Japan) to identify cell attachments on their surface. The osteoblasts that adhered to the samples were fixed in a solution of 3% glutaraldehyde (50 wt.% in H_2_O, CAS#111-30-8, Sigma-Aldrich), 0.1 mol/L sodium cacodylate (CAS#6131-99-3, Sigma-Aldrich), and 0.1 mol/L sucrose (CAS#57-50-1, Sigma-Aldrich) for 45 min. Samples were soaked for 10 min in a buffer solution of 0.1 mol/L sodium cacodylate (CAS#6131-99-3, Sigma-Aldrich) and 0.1 mol/L sucrose (CAS#57-50-1, Sigma-Aldrich). Sample surfaces and cells were processed in serial ethanol dehydrations for 10 min each and dehydrated in hexamethyldisilazane (HDMS, CAS# 999-97-3, Sigma-Aldrich) before stored in a desiccator until SEM imaging. Titanium sheets were then sputter-coated with a palladium–gold alloy (Polaron SC 7620 Sputter Coater, Quorum Technologies, Laughton, East Sussex, UK) with a thickness of 10 nm (10–15 mA, under a vacuum of 130 mTorr). After this, the SEM was operated at 5 kV, spot 3 to 6.

The quantitative data were shown as the means ± standard deviations. The statistical differences were calculated using one-way ANOVA and Tukey’s test (GraphPad Prism 9 Software, Inc, San Diego, CA, USA). A *p*-value of ≤0.05 was considered statistically significant.

## 3. Results

### 3.1. Cytotoxicity and Cell Proliferation

In order to determine whether the nanotubes demonstrated biocompatibility, cytotoxicity was evaluated by the CellTiter-Blue assay. After 1 day and 3 days of human osteoblasts cultivation on the samples, no obvious cytotoxicity was observed, as evidenced by the absorbance of the cells cultured on nanotubes, being comparable to the control group (*p* = 0.287) (Figure 2). In addition, after seven days, we observed higher cell proliferation on the nanotubes samples (*p* = 0.3756).

### 3.2. Cell Attachment

The human osteoblasts were observed by SEM on all experimental samples after 1 day in culture and cells were distributed on their surfaces (Figure 3).

In Figure 4A,B, the cell morphology appeared oval-shaped and flattened on the nanotube surface. Additionally, SEM images showed the high interaction of cell extensions with the nanotube surfaces (Figure 4C,D).

### 3.3. Biofilm Adhesion

SEM images demonstrated low biofilm adhesion for TiO_2_ nanotube sheets after 30 days cultivation (Figure 5). The lowest amount of biofim adhesion was found in TiO_2_ nanotubes coated with SiC.

### 3.4. Biofilm Coverage

After 30 days, the surface adherence of *P. gingivalis* was less on the TiO_2_ nanotube sheets, showing a biofilm coverage lower than 8% (Figure 6). For *T. denticola*, the TiO_2_ nanotube sheets also showed reduced biofilm adherence (9% and 7%). The TiO_2_ nanotube sheets coated with SiC presented low biofim surface adherence for *T. forsythia* and *T. denticola*.

### 3.5. Analysis of Nanotube Surface by SEM after Bacterial Inoculation

Figure 7 demonstrates the change in surface roughness of Ti non-coated and Ti coated nanotubes sheets after bacterial inoculation and removal. Ti non-coated nanotubes sheet (Figure 7B) showed pitting and surface breakdown, while these were not evident with the coated samples (Figure 7C).

## 4. Discussion

Although titanium has already been used for several purposes [27], the toxicity of this metal has not been addressed in detail, considering the direct and indirect consequences and acute and chronic toxicity [28]. In the present study, no cell cytotoxicity was found after 24 h in nanotube titanium sheets (100 nm and 150 nm) showing potential biocompatibility, which is a favorable property for osseintegration. Huo et al. [12] verified that titanium nanotube implants promoted high osteogenic activity in osteoblasts in vitro.

In addition, titanium with nanotubes promoted cell proliferation on osteoblasts after seven days. Capelatto et al. [19] verified that titanium nanotube samples presented a significant effect on the number of cells present, showing an increase of cells from day 1 to day 7. The cell proliferation assay results showed that titanium nanotubes (100 nm) could promote high cell proliferation compared with control Ti [29]. Furthermore, Ti nanotube alloys can induce cell growth and biocompatibility in stem cells after one and seven days in culture [10].

Titanium surfaces should be suitable for cell attachment to achieve proper and early osseointegration. In this study, we observed cells on the nanotube titanium sheets (100 nm or 150 nm) by SEM. In addition, titanium nanotubes with a diameter of 100 nm promoted cell attachment of human gingival fibroblasts [29]. Kumeria et al. [13] reported that an advanced biopolymer-coated drug-releasing titanium nanotubes implant enhanced osteoblast adhesion. Si-doped TiO_2_ nanotubes induced the osteogenic differentiation of osteoblastic cells and improved bone-titanium integration [30]. The Zn-ZrO_2_/TiO_2_ porous coatings on the Ti6Al4V surface can promote a relatively stable and biofriendly interface for cell adhesion and growth, thus highlighting favorable cytocompatibility [8].

To further improve the material’s longevity, thus preventing peri-implantitis, one of the significant strategies focuses on avoiding bacterial adhesion on dental implant material [1,31,32,33]. Although complete prevention of bacterial adhesion on biomaterials is complicated, various efforts have been developed to effectively minimize and control the bacterial adhesion on biomaterial surfaces [15,20,33,34]. In this way, new coatings have been developed on the surface of titanium to decrease the biofilm [6,7,15,35].

Analysis of the nanotube surfaces with SEM before and after bacterial inoculation demonstrated the possibility of nanotubes corroding with exposure to bacteria. SEM reveals blunting of nanotube edges after exposure, compared with the reference nanotube surfaces before exposure. SiC-coated nanotubes remained the same before and after exposure to bacteria with no evidence of corrosion on the surface. This effect of SiC is consistent with previous studies. Camargo et al. [35] showed that coatings based on SiC and TiN applied on Ti surfaces were effective against *P. gingivalis* in vitro. Titanium nanotubes (100 nm) could also decrease the adhesion of *P. gingivalis* to the transgingival area, facilitating the attachment of soft tissue, inhibiting the adhesion of bacterial biofilm, and preserving the crestal bone [29]. In this study, we observed that biofilm adhesion was low in titanium nanotube sheets. SiC coatings for dental applications were also reported to exhibit good corrosion resistance [23,36]. Hsu et al. [37] established that the ceramic samples coated with SiC were not corroded compared to the non-coated, suggesting the protective quality of these coatings. Here, we observed that the surface of non-coated nanotube sheets presented irregularities similar to corrosion while the SiC coating on titanium nanotubes presented smoother surfaces.

The findings of this study are promising in that Ti nanotubes demonstrated good cell proliferation for bone growth and SiC-coated nanotubes prevented adhesion and colonization of bacteria on the surface. However, the results of this study remain limited, and additional studies are necessary in vivo to determine their influence on clinical outcomes.

## 5. Conclusions

According to the limitations of this in vitro study, Ti nanotubes can promote osteoblast proliferation and reduce biofilm adhesion, which is essential to achieve consistent osteointegration. In addition, the SiC coating demonstrated a promising effect in minimizing Ti corrosion from bacterial biofilms, which is essential for the longevity of dental implants.

Thus, in vivo studies should be performed to better understand the advantages of Ti nanotubes on dental implants.

## Figures and Tables

**Figure 1 materials-14-04357-f001:**
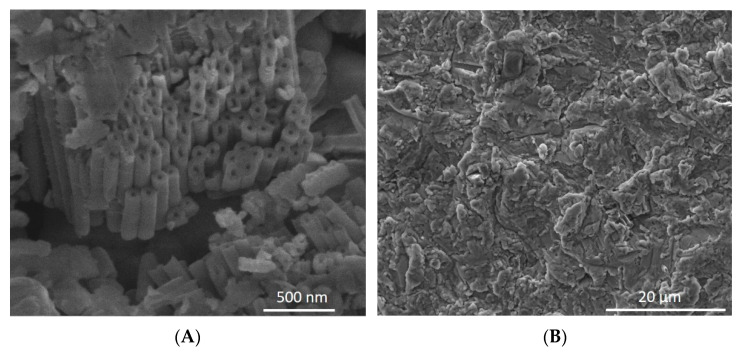
SEM images of dental implant showing nanotubes on the surface (**A**). The image (**B**) shows dental implant coated with SiC.

**Figure 2 materials-14-04357-f002:**
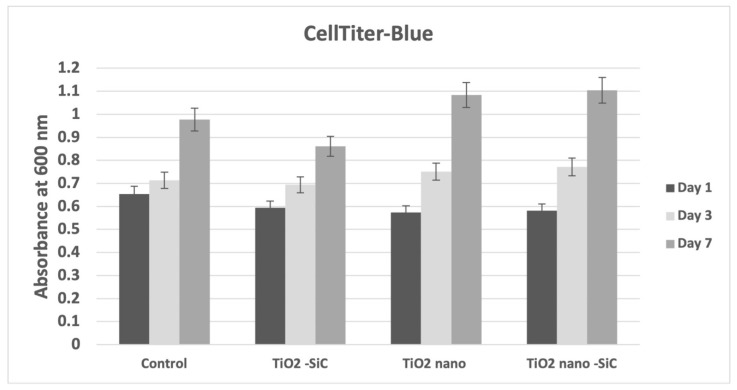
Cytotoxicity of the control (only cells) and titanium oxide coated with silicon carbide (TiO_2_-SiC), titanium oxide nanotubes (TiO_2_ nano), titanium oxide nanotubes coated with silicon carbide (TiO_2_ nano-SiC) groups after 24 h of human osteoblasts (Nhost) culture assessed by CellTiter-Blue absorbance.

**Figure 3 materials-14-04357-f003:**

SEM images showing human osteoblasts on titanium sheets after 1-day culture. (**A**)Titanium oxide coated with silicon carbide (TiO_2_-SiC), (**B**) Titanium oxide nanotubes (TiO_2_ nano), (**C**)Titanium oxide nanotubes coated with silicon carbide (TiO_2_ nano-SiC).

**Figure 4 materials-14-04357-f004:**
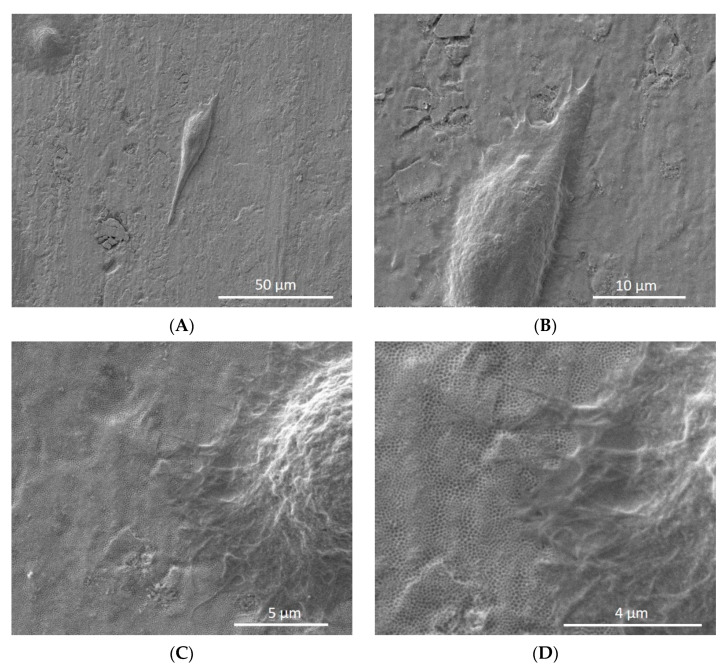
Human osteoblasts on the surface of nanotube sheets [titanium oxide nanotubes (TiO_2_ nano) (**A**,**B**) and titanium oxide nanotubes coated with silicon carbide (TiO_2_ nano-SiC) (**C**,**D**)].(**A**,**B**)—cell elongated on the surface of nanotube sheets, (**C**,**D**)—cell in intimate contact with the nanotube sheet.

**Figure 5 materials-14-04357-f005:**
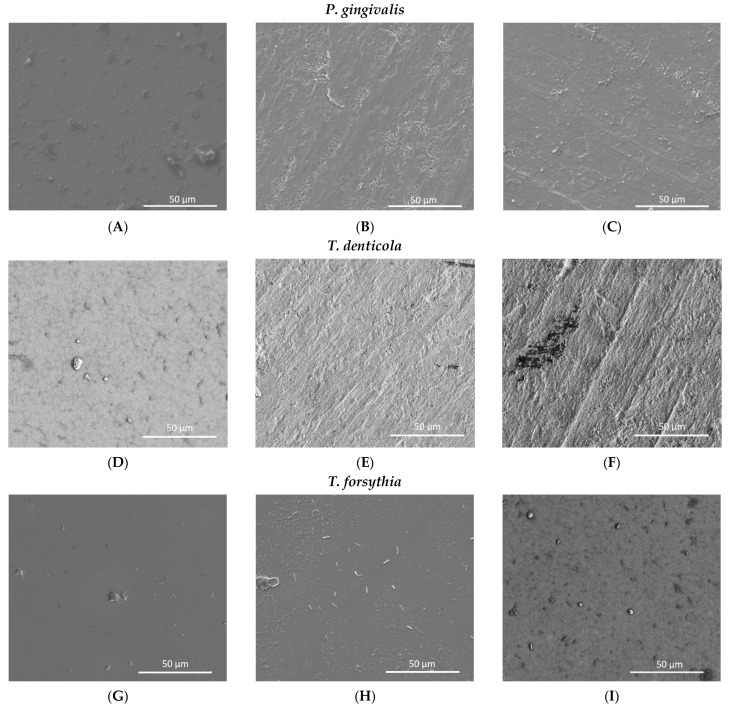
SEM adhesion of monobacterial biofilm of *P. gingivalis, T. denticola,* and *T. forsythia* on titanium oxide coated with silicon carbide (TiO_2_-SiC) (**A,D,G**), titanium oxide nanotubes (TiO_2_ nano) (**B,E,H**), and titanium oxide nanotubes coated with silicon carbide (TiO_2_ nano-SiC) (**C,F,I**) groups.

**Figure 6 materials-14-04357-f006:**
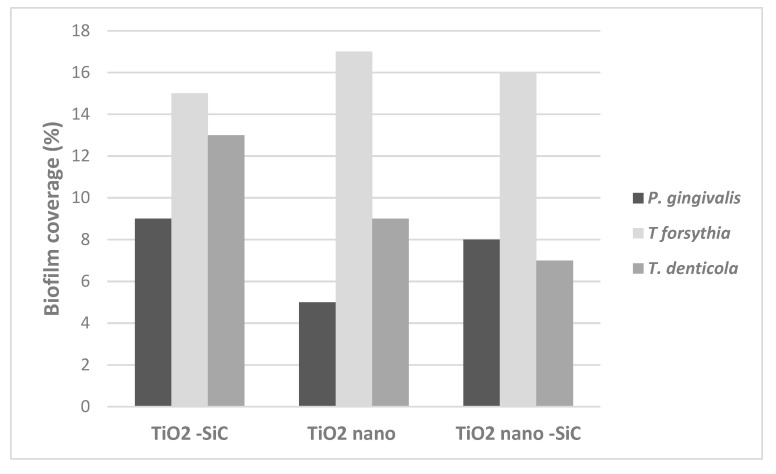
Monobacterial biofilm coverage on the surface of titanium oxide silicon coated with silicon carbide (TiO_2_-SiC), titanium oxide silicon nanotubes (TiO_2_ nano), and titanium oxide silicon nanotubes coated with silicon carbide (TiO_2_ nano-SiC) groups.

**Figure 7 materials-14-04357-f007:**
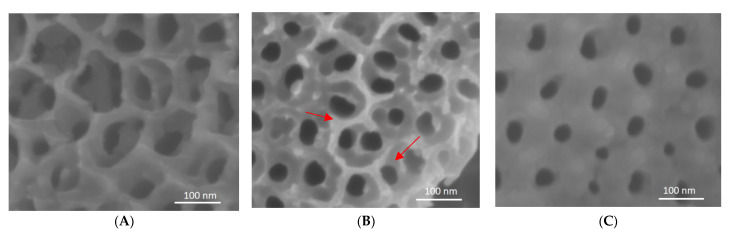
SEM images of reference ((**A**)—no bacteria, non-coated) and Ti nanotubes sheet non-coated (**B**) and coated (**C**) after 30 days in contact with *P. gingivalis*. Arrows showing surface breakdown.

## Data Availability

All data included in this study are available upon request by contact with the corresponding author.

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
