# Peer review of "Nanostructured Surfaces to Promote Osteoblast Proliferation and Minimize Bacterial Adhesion on Titanium"

_materials, 2021, doi:10.3390/ma14164357_

Round 1

Reviewer 1 Report

The manuscript is well written but I have few suggestions.

  1. The SEM graphs are needed to be converted into high resolution.
  2. Figure 2. is not appropriately reported instead of using color scheme author should provide this graph with black, white and shaded columns.
  3. Figure 4 needed to be properly described in detail in the text.
  4. Discussion is very small having few referecnes. I think there is need of improvement as well
  5. Conclusion is insufficient. please write down more.

Reviewer 2 Report

Camargo et al. studied the potential of titanium nanotubes and silicon carbide coated nanotubes to promote human osteoblast proliferation, to reduce monomicrobial biofilm adhesion. Below are my comments,

Major comments:

  • Figure 1: Instead of showing two SEM images (A & B) for titanium nanotubes, one would be enough.
  • The authors only provided SEM images of cells after one day of culture. Please provide similar SEM images after three and seven days of culture.
  • Figure 4: Please indicate in the figure legend which SEM images are for TiO2 nano and TiO2 nano-SiC.
  • Please clarify which bacteria was used for figure 5 and provide the corresponding SEM images for the rest two bacteria.
  • For the general reader, please discuss why these three bacteria were selected for this study.
  • Figure 7: Please indicate in the text and figure legend which bacteria was used for this figure.
  • In the current manuscript, the authors comment on the cytotoxicity only after 1-7 days. This short duration will have little relevance in the real-life scenario.

Minor points:

  • Figure 1, 3-5, 7: Specifications on the images are hard to read. Increasing font size will help the visibility.
  • Please indicate in the legends of figure 3 that the SEM images are after one day of culture.

Reviewer 3 Report

Dear Authors,

The topic is generally very interesting - considering the use of nanomaterials in dental prosthetics. However, the article in this form cannot be accepted for publication because it has serious shortcomings. I suggest that the following deficiencies be corrected:

- in the methods section you have no description of the chemical reagents you use, the numbers, CAS, the preparation is very poorly described,

- the literature you use is a bit new, there are no new facts about proliferation,

- your research methodology is very poor - present only results from SEM and cytotoxicity tests, why? What about for example scratch test for nanomaterials, XPS or other chemical tests, what about ICP in SBF solutions for new materials? In general, these are serious shortcomings that give a bad indication of a research workshop,

- and there is also an inaccuracy in the idea itself, because these are not nanotubes with titanium but titania - and strictly speaking these are materials in nano form applied on TiO2, this is a big inaccuracy and it must be corrected because you are misleading your readers.

Without correcting all the above flaws I do not recommend the article for acceptance. Please correct them.

Best Wishes

Reviewer

Round 2

Reviewer 2 Report

Major points

  • The authors checked the cell adhesion by SEM only after 1 day of culture and they do not have any data on the cell adhesion after three or seven days. This duration is too short and certainly not sufficient to claim that Ti nanotubes promoted cell adhesion on their surface. The authors require to provide additional data at least after three and seven days.

Minor points

  • Figure 4D appears to be zoomed out for a portion of figure 4C. Is this correct? In that case, the scale bar for figure 4D is not correct.

Author Response

Thank you for the suggestions.

You can find the answers in the attached file. 
